# Genetic Variation of the Serine Acetyltransferase Gene Family for Sulfur Assimilation in Maize

**DOI:** 10.3390/genes12030437

**Published:** 2021-03-19

**Authors:** Zhixuan Zhao, Shuai Li, Chen Ji, Yong Zhou, Changsheng Li, Wenqin Wang

**Affiliations:** 1School of Agriculture and Biology, Shanghai Jiao Tong University, Shanghai 200240, China; zzx0908@sjtu.edu.cn (Z.Z.); ls2019@sjtu.edu.cn (S.L.); 2National Key Laboratory of Plant Molecular Genetics, CAS Center for Excellence in Molecular Plant Sciences, Institute of Plant Physiology & Ecology, Shanghai Institutes for Biological Sciences, Chinese Academy of Sciences, Shanghai 200032, China; jichen@cemps.ac.cn (C.J.); yongzhou20160918@gmail.com (Y.Z.); csli@cemps.ac.cn (C.L.)

**Keywords:** methionine, cysteine, transposon, nested association mapping population

## Abstract

Improving sulfur assimilation in maize kernels is essential due to humans and animals’ inability to synthesize methionine. Serine acetyltransferase (SAT) is a critical enzyme that controls cystine biosynthesis in plants. In this study, all SAT gene members were genome-wide characterized by using a sequence homology search. The RNA-seq quantification indicates that they are highly expressed in leaves, other than root and seeds, consistent with their biological functions in sulfur assimilation. With the recently released 25 genomes of nested association mapping (NAM) founders representing the diverse maize stock, we had the opportunity to investigate the SAT genetic variation comprehensively. The abundant transposon insertions into SAT genes indicate their driving power in terms of gene structure and genome evolution. We found that the transposon insertion into exons could change SAT gene transcription, whereas there was no significant correlation between transposable element (TE) insertion into introns and their gene expression, indicating that other regulatory elements such as promoters could also be involved. Understanding the SAT gene structure, gene expression and genetic variation involved in natural selection and species adaption could precisely guide genetic engineering to manipulate sulfur assimilation in maize and to improve nutritional quality.

## 1. Introduction

Maize is one of the most important crops in terms of its high yield and broadly derived commodities. Zeins are the major storage proteins in maize endosperm, but they are devoid of the essential amino acids lysine and tryptophan. The most abundant zein proteins are α- and γ-zeins. They also lack another essential amino acid of methionine (Met), a sulfur-containing amino acid [1]. The deficiency of lysine and tryptophan in a corn-based diet could be overcome by supplementing soybean, which is rich in the two amino acids but is still devoid of Met. Therefore, Met in a corn–soybean formula feed is enhanced by adding chemically synthesized methionine, which increases the food supply cost. Although scientists tried to improve the S-amino acids by overexpressing genes encoding Met- or Cys-rich proteins, an unchanged total content of sulfur was achieved due to the mutual re-allocation between Met and Cys, indicating a limited sulfur availability or reduction in maize [2].

A practical approach to manipulate the sulfur-containing amino acid content in kernels is to increase the source assimilation. Sulfur is an essential macronutrient and is stored as organic sulfur compounds in plants. They can reduce absorbed sulfate from roots, assimilate into cysteine in leaves and further metabolize into methionine, glutathione and other compounds [3]. The primary committing steps for Cys synthesis include sulfate reduction by 5′-phosphosulfate reductase (APR) and O-acetyl-L-serine (OAS) formation by serine acetyltransferase (SAT) [4]. APR controls the flow of inorganic sulfur into cysteine by producing H2S. SAT could generate the activated carbon backbone for Cys biosynthesis from L-Ser and acetyl-CoA’s catalysis into OAS [5]. Overexpression of APR and SAT resulted in an enhanced Met accumulation in seeds and the total sulfur-containing amino acids. The transgenic maize kernels showed improved nutritional quality and could increase the growth rate of the fed chicken [6,7].

There is a considerable variation of methionine levels in natural maize inbred with a range of insufficient to sufficient animal feed levels, suggesting that screening alleles of APR and SAT genes in maize is feasible for genetic improvement. BSSS53 contains a significantly higher Met level than other inbred lines such as Mo17 and W64A due to the elevated accumulation of 10-kDa δ-zein, which includes 23% of Met residues in the protein sequence. When BSSS53 was crossed with other lines, the Met levels were increased by 23–43% in the backcross lines [8], indicating that BSSS53 is a promising donor to improve the Met content. Still, a more in-depth study to investigate the genetic variation in maize populations will promote an understanding of the mechanism of biosynthesis of sulfur amino acids. The SAT gene family controls the crucial step of sulfur assimilation and shows promise to improve sulfur content using transgenic technology. The exploration of genetic variation and expression patterns for the SAT gene family would provide sufficient genetic resources and superior alleles to further manipulate maize’s sulfur content.

The recently assembled and annotated maize genomes including B73 [9], Mo17 [10] and K0326Y [11] revealed that transposons amounting to ~83% are largely responsible for extensive variation in intergenic and genic regions. Studying the composition, function and transcriptional regulation of transposons in or near genes would expediate our understanding of the relationship between genetic diversity and phenotypic variation. There are increasing examples demonstrating that transposable element (TE) insertion could evolve novel functions and affect their expression by providing regulatory or silencing signals in different crops or plant species. The transposable element inserted upstream of the teosinte branched1 (tb1) gene could enhance the apical dominance with the increased gene expression, partially domesticating maize from its progenitor of teosinte [12]. Retrotransposon insertion into an anthocyanin biosynthetic gene, VvmybA1, in grape affected cis-regulatory variation and led to a berry pigmentation change and created a new phenotype [13].

The 25 nested association mapping (NAM) parental lines (B97, CML52, CML69, CML103, CML228, CML247, CML277, CML322, CML333, HP301, Il14H, Ki3, Ki11, Ky21, M37W, M162W, Mo18W, Ms71, NC350, NC358, Oh7B, Oh43, P39, Tx303 and Tzi8) representing diverse maize lines in modern breeding [14] were selected for sequencing and deciphering maize genetic diversity [15]. They became a set of essential resources for the maize community to answer critical questions about how structural variations determine the phenotypic traits and adaption to the different environments. Along with the DNA sequencing era, the most extensively studied maize NAM parental lines were recently sequenced and assembled (https://nam-genomes.org, accessed on 15 March 2020). Furthermore, an expanded maize gene expression atlas based on RNA sequencing, including a time course of stalk, leaf, flower, seed and root, was generated [16] and deposited in the MaizeGDB database (https://qteller.maizegdb.org, accessed on 12 July 2019). With the availability of 25 genomes and the spatiotemporal transcriptomic atlas, we had the opportunity to identify all members of the SAT gene family and to investigate the correlation of structural variations of SAT alleles with their gene expression and enzyme activity. Here, we present a comprehensive genetic characterization of the SAT family members in maize B73 and 25 NAM founders. The further molecular genetic analysis of SATs involving gene expression and transposon insertion paves the way to precisely manipulate sulfur assimilation in maize.

## 2. Materials and Methods

### 2.1. Identification of the SAT Gene Family

The genome of *Zea mays* L. (GenBank accession number: SAMN04296295, Zm-B73-REFERENCE-GRAMENE-4.0) was used as a reference to identify the SAT genes. Basic Local Alignment Search Tool (BLAST) was utilized to search the SAT genes based on sequence similarity using previously identified SAT proteins from model plants of *Arabidopsis* and rice. The BLAST parameters were defined with an e-value threshold of 1E−10 and a retrieval of 10 best hits. In addition, the candidates were validated according to the existence of common motifs of SATase_N and Hexapep_C. The hidden Markov model (HMM) files for SATase_N domain (PF06426) and Hexapep_C (PF00132) were downloaded from the Pfam protein family database (http://pfam.sanger.ac.uk/, accessed on 2 February 2020). HMMER 3.0 was used to search the conserved motifs by using the default parameters. The genome of *Arabidopsis* was obtained from the Arabidopsis Information Resource (TAIR) (http://www.Arabidopsis.org, accessed on 15 March 2020). The 25 genomes of the nested associated mapping (NAM) founders as representatives of modern maize diversity generated by the NAM Consortium Group were retrieved from GenBank (BioProject ID: PRJEB31061). The SAT alleles in the NAM population were identified with the alignment of B73 cDNA to genomes using the Exonerate est2genome model [17]. The multiple SAT sequences from maize B73 and *Arabidopsis* were aligned using MUSCLE (Multiple Sequence Comparison by Log-Expectation: https://www.ebi.ac.uk/Tools/msa/muscle, accessed on 13 April 2020) with default parameters. Transposable elements were identified using the CENSOR tool, which searched query sequences against a reference collection of repeats (https://www.girinst.org/censor/index.php, accessed on 18 November 2020) [18]. The phylogenetic tree was drawn using concatenated SAT sequences and the MEGA X (Molecular Evolutionary Genetics Analysis: https://www.megasoftware.net, accessed on 18 November 2020) program [19].

### 2.2. Annotation of the SAT Genes

The SAT alleles were manually annotated with the online tool FGENESH_C (HMM plus similar cDNA-based gene structure prediction, http://www.softberry.com, accessed on 13 April 2020) [20]. The deduced amino acid sequences in SAT domains were manually double-checked using GeneDoc software (https://github.com/karlnicholas/GeneDoc, accessed on 13 April 2020). The exon–intron organization of SAT genes was determined by comparing predicted coding sequences with their corresponding full-length sequences using the online program Gene Structure Display Server (GSDS: http://gsds.cbi.pku.edu.cn, accessed on 1 May 2020). The conserved motifs in SAT proteins were characterized using the MEME online program (http://meme.nbcr.net/meme/intro.html, accessed on 24 May 2020).

### 2.3. Transcriptome Analysis

To identify the spatiotemporal expression of SATs, the RNA-Seq dataset was used to quantify SAT gene expression. The raw data were initially sequenced and analyzed by Kaeppler lab from 79 different replicated B73 samples to build a comprehensive transcriptome dataset that could provide a powerful tool for understanding maize development, physiology and phenotypic diversity [16]. The dataset had an average of 20.5 million mapped reads per biological replicate, allowing for gene detection even with low transcription. Basically, all RNA-seq reads were mapped by Bowtie, and fragments per kilobase of gene model per million reads mapped (FPKM) were counted by Cufflinks [16,21]. Here, we directly retrieved the SAT gene expression with FPKM values and standard errors from the Kaeppler lab, from which we could further visualize the spatiotemporal gene expression and infer their molecular functions. To verify the effects of TE insertions on the associated SAT genes, we carried out a qRT-PCR analysis to quantify their gene expression using three biological replicates and the gene actin as an internal control. A panel of inbred maize lines (B73, B97, CML103, Ki11 and Ky21) were grown at Sanya, Hainan, China (E 109°42′, 18°12′) in 2020. The sixth leaves at the R3 stage were sampled, frozen in liquid nitrogen and stored later at −80 °C. The total RNA was extracted using an RNA Easy Fast Plant Tissue RNA Rapid Extraction Kit (DP452, TIANGEN, Beijing, China). AN amount of 5 ug total RNA was used for reverse transcription with the Superscript III first-strand kit (Invitrogen). The primers for qRT-PCR were as follows (SAT1-F: GTCCTCGGAAACGTCAGGAT; SAT1-R: CAAATGATGTAGTCCGACCACTG; SAT3-F: TATTATGGAACCAGGGACGCAAA, SAT3-R: GAAACCCAGTTTCCAACTACAGC; SAT4-F: TTGGTAACGTGAAAATTGGTGCC; SAT4-R: TTGACGTATGAAGGACGTGTGAT). The resulting cDNA was diluted to 10 ng/ul for each sample. The real-time PCR was performed with SYBR Green Premix Pro Taq HS qPCR Kit AG11701 (Accurate Biology, Changsha, China) and a Bio-Rad CFX Connect thermocycler (Bio-Rad, Hercules, CA, USA). Statistically significant differences in gene expression levels were analyzed using the ANOVA test (*p*-value of 0.05).

### 2.4. Phylogenetic Tree Analyses

To study the SAT gene evolution, we performed a phylogenetic analysis using the identified 50 SAT proteins across cyanobacteria to seed plants, including a cyanobacterium of *Synechococcus* sp. WH 7803, two species of *Galdieria sulphuraria* and *Chondrus crispus* from red algae, three species of *Micromonas* sp. RCC299, *Micromonas pusilla* CCMP1545 and *Chlamydomonas reinhardtii* in green algae, one species of *Mpolymorpha* in moss, a fern of *Selaginella moellendorffii*, a gymnosperm of *Picea abies*, a basal angiosperm of *Amborella trichopoda*, four species (*Brachypodium distachyon*, sorghum, rice and maize) in monocots and two species (*Vitis vinifera* and *Arabidopsis*) in dicots [22].

### 2.5. Data Source

The reference B73 genome (Zm-B73-REFERENCE-GRAMENE-4.0) was download from MaizeGDB (https://www.maizegdb.org/, accessed on 12 July 2019). The genomic data of 25 NAM founders for identifying genetic variation were obtained at the NAM genomes website (https://nam-genomes.org/, accessed on 15 March 2020). The quantitative RNA-seq data were derived from Kaeppler’s lab [16]. The genome data of *Chondrus crispus* and *Galdieria sulphuraria* were download from Ensembl (http://plants.ensembl.org/index.html, accessed on 12 July 2019). The genome data of *Synechococcus* sp. WH7803 (GenBank accession: GCA_000063505.1), *Vitis vinifera* (GenBank accession: GCA_000003745.2) and *Sorghum bicolor* (GenBank accession: GCA_000003195.3) were downloaded from NCBI (https://www.ncbi.nlm.nih.gov/, accessed on 12 July 2019). The genome data of *Picea abies* were obtained from Plantgenie (ftp://plantgenie.org/Data/ConGenIE/Picea_abies/, accessed on 12 July 2019). The genomes of other species (*Chlamydomonas reinhardtii*, *Micromonas pusilla* CCMP1545, *Micromonas* sp. RCC299, *Mpolymorpha*, *Selaginella moellendorffii*, *Amborella trichopoda* and *Brachypodium distachyon*) were retrieved from Phytozome (https://phytozome.jgi.doe.gov/pz/portal.html, accessed on 12 July 2019).

## 3. Results

### 3.1. Identification of SAT Proteins in Maize B73 Inbred Line

A total of four SAT gene family members were characterized in the B73 genome by homology searches. They were located on chromosomes 1, 6 and 8. They were designated according to their protein sequence lengths (SAT1—Zm00001d011735; SAT2—Zm00001d028154; SAT3—Zm00001d027536; SAT4—Zm00001d038737) (Figure 1). SAT1 and SAT2 have a single exon, while SAT3 contains nine and SAT4 three exons. The coding sequences range from 933 to 1056 bp and encode SAT proteins of 310–351 amino acids with a molecular mass of 32.4–37.4 kDa (Table 1). All members carry an N-terminal serine acetyltransferase motif (SATase_N) and a C-terminal hexapeptide repeat domain (Hexapep_C). The SATase_N is involved in enzymatic activity and the Hexapep_C is reported to be critical for the formation of the SAT–cysteine synthase complex [23]. The gene of Zm00001d011738 was excluded due to the missing SATase_N domain. There are five copies in *Arabidopsis* [24] and four in *Sorghum* [25] sharing both motifs of SATase_N and Hexapep_C. The multiple sequence alignment of SAT proteins from maize and *Arabidopsis* demonstrated a high similarity in the middle region of these proteins, whereas the N- and C-terminal regions were variable (Figure 2).

### 3.2. Expressional Profiling of the Maize SAT Genes

To investigate the SAT spatiotemporal transcriptome profiles, we took advantage of the expanded maize gene expression atlas based on RNA sequencing from Kaeppler’s lab [16] in the B73 inbred line (Figure 3). The samples were derived from different developmental stages of maize seeds, roots and leaves with three biological replicates. The mean values and standard errors of the SAT gene expression were directly retrieved from the Kaeppler lab [16]. We found that the SAT genes were more highly expressed in leaves than in the seed and root tissues (Figure 3), consistent with their function in Cys and Met biosynthesis in leaves, where SAT is a committing enzyme for sulfur assimilation. In addition, SAT1 was the most expressed gene in leaves among all members and increased significantly at the stage of V9 leaf, but it was not expressed in seeds and roots. SAT2 was transcribed less than SAT1 in leaves and was not detected in seeds and roots. SAT3 was constitutively expressed in all stages and tissues, while SAT4 was also mainly expressed in leaves, but with low expression in other organs (Figure 3). The SAT1 and SAT4 in leaves contributed to 68–90% of total gene expression, while SAT2 and SAT3 amounted to 10–32% of the remaining abundance at the stage of V9, VT and R2 (Figure 3). The elevated gene expression of SAT1 and SAT4 in the late vegetative and reproduction stages is consistent with their role in Cys biosynthesis for the coming process of seed storage.

### 3.3. Structural Variation of SAT Genes in NAM Population

The single genome reference of B73 could not provide sufficient genetic architectural information on the SAT gene family due to the considerable natural variation in maize inbred lines. NAM founders represent a set of diverse inbred lines that are publicly available and used extensively in the maize community to investigate the genetic basis of complex traits [14]. The 25 parent inbred lines (called NAM founders) were selected from the maize stock to maximize the genetic diversity, sequenced using the PacBio third-generation platform and released to the public recently (https://nam-genomes.org, accessed on 15 March 2020). We had the unprecedented opportunity to study the population structure of the SAT alleles in maize. Previous studies showed that the gene content differs by more than 5% across lines and half of the functional genetic information was derived from highly variable intergenic sequences [14]. Here, we focused on capturing natural variation only inside the SAT gene regions that were tightly correlated with gene expression and protein translation.

The SAT1 gene (ID: SAT1—Zm0000d011735) contains a single exon with a size of 933 bp that encodes a protein of 311 amino acids. We identified 15 polymorphic SNPs (Single Nucleotide Polymorphism) and a 12,977-bp CACTA transposon insertion at 717 bp after the start codon in the Ki11 genome (Appendix A), which was 91% identical to the consensus from the repeat database with a 3-bp target site duplication (TSD, 5′-ATC-3’) and 19-bp imperfect terminal inverted repeats (TIRs) (Figure 4a). The transposon also carries a partial gene (HARBI1) of 2297 bp encoding a nuclease, consistent with the previous findings that CACTA transposable elements could capture gene fragments in maize [26]. A small long terminal repeat (LTR) retrotransposon with 536 bp was identified at 825 bp after the start codon in SAT1 in Ky21 (Figure 4a). Further analysis revealed that the transposon had typical characteristics of terminal-repeat retrotransposons in miniature, called a Wukong element [27], with a 5-bp TSD (GGGAA). There were two 232-bp long terminal repeats (LTRs) and a 67-bp internal domain with two conserved motifs (PBS of ATTGGTATCAGAGCCA and PPT of GAGGGGGAGAT) that were required for reverse transcription in the Wukong element. The transposon insertion into the exon changed the SAT1 gene structure and resulted in a premature stop codon in the coding sequence. The quantitative real-time PCR verified that the transposon insertion into coding regions inhibited SAT1 gene expression in Ki11 and Ky21. As expected, they were not affected in B73, B97 and CML103, where no TE insertion in exons occurred (Figure 5a).

The SAT2 gene (ID: SAT2—Zm00001d028154) was relatively more conserved than other members with only 13 SNPs in all NAM founders and a small 12-bp deletion in Ky21 (Appendix A). The SAT3 gene (ID: SAT3—Zm00001d027536) contained a large sixth intron, exhibiting a hotspot for transposon insertion. We found a 2066-bp Copia insertion upstream and a 2764-bp L1 non-LTR retrotransposon in the middle of the sixth intron in CML103, CML228, NC358 and P39. There was a 3142-bp Gypsy inserted in Oh7B and HP301, and an 1866-bp Copia in CML52, II14H and Ky21 (Figure 4b). The number of polymorphic SNPs in SAT3 was as high as 409, possibly due to the multiple introns and their large sizes (Appendix A). The SAT4 gene (ID: SAT4—Zm00001d038737) structures were conserved in the first and second exons in B73, B97, Il14H, Ki11 and P39, whose alleles also shared an 848-bp DNA transposon of hAT. There was a 9524-bp Copia inserted only in Ki11 (Figure 4c). In total, 13 SNPs were found in the third exon in all NAM founders (Appendix A). The SAT3 gene expressions were reduced in Ky21 and increased in B97 and CML103 compared with B73. There was no significant difference for SAT3 and SAT4 between B73 and Ki11 (Figure 5b). SAT4 showed improved transcription in CML103, Ky21 and B97 in comparison with B73 (Figure 5c). However, we did not find a significant correlation between intron variation and gene expression, indicating that other elements such as promoters may contribute to gene regulation beyond introns.

We then combined all the SAT members to create a phylogenetic tree, wherein the NAM founders were clustered into three branches composing 13, 9 and 4 lines, respectively. We found that the SAT clustering did not comply with their geographic locations, whereas the breeding groups from temperate, flint and intermediate maize were mixed (Figure 6). Thus, the SAT gene family was not an optimal marker to trace inbred maize lines regarding their geographical origin.

### 3.4. Evolutionary Analysis of SAT Genes

To study the SAT genes’ origin and evolutionary process, a phylogenetic tree was drawn and the 50 SAT genes were classified into six clades (I, II, III, IV, V and VI). The five clades (I, II, III, IV and V) included 27 SAT proteins from red algae (3), green algae (8), moss (2), fern (1) and seed plant (13), showing that the clades with two maize SAT copies (SAT2 and SAT3) were ancient and might originate from cyanobacteria. The VI clade contained 22 SAT proteins from moss (1), fern (2) and seed plant (19), indicating that the clade with the other two maize SAT copies (SAT1 and SAT4) was recently evolved and might originate from moss (Figure 7).

## 4. Discussion

### 4.1. Functional Redundancy of the SAT Gene Family

Homology searching of the B73 genome resulted in the identification of four members of the SAT gene family in maize, compared to five in *Arabidopsis* [23] and four in *Vitis* [28]. It is known that the SAT proteins are basically conserved and function in all subcellular compartments among vascular plants [28,29]. The multiple SAT family members might have acquired an adaptive advantage by allowing differential regulation in response to different endogenous and environmental stimulations. Gene redundancy might serve as an invaluable source of evolutionary innovations against deleterious mutations. The loss of function of SAT1 due to the transposon insertion into exons does not change cystine and methionine contents [30], suggesting that other functional SAT members could compensate for the loss of SAT1 (Figure 4). The single SAT knockout mutants in *Arabidopsis* did not show any visible phenotypic changes, while quintuple mutants were lethal, indicating their essential but redundant enzymatic function [5]. The spatiotemporal expression of maize SATs showed that their transcriptions are regulated in response to tissue types, developmental stages and environmental stimuli (Figure 3). It has also been generally accepted that SAT activities can be adjusted by sulfate supply and internal cysteine demand. AtSAT5 in *Arabidopsis* is induced under heavy metal stress, allowing the plant to withstand metal toxicity by increasing the sulfate assimilation rate [24]. Upon sulfate depletion in *Vitis*, AtSAT2-1 was substantially increased at the transcriptional level to overcome sulfate limitation [28]. With the rapid development of genetic engineering techniques, the bacterial adenosine 5′-phosphosulfate reductase (APS) and bacterial SAT were transformed into maize to improve sulfur storage. Still, it is a compromised method to skip complex regulatory control in higher plants [6,7]. Therefore, a better understanding of the gene structure, expression pattern and genetic regulation of the SAT gene family will provide an effective approach to substantially increase the Cys and Met contents in future maize breeding.

### 4.2. Transposable Elements and Gene Evolution

Repetitive DNA sequences account for at least 85% of the maize genome [10] and are a major driver of genome and gene evolution [31]. They reshaped the genomes with their dynamic activity, which provides genetic variation to influence the gene function. The evolutionary relationship of genes can be analyzed by constructing evolutionary trees. However, the SAT gene family was not an optimal marker to trace maize inbred lines in terms of their geographical origin. TEs can change the gene structure through insertion into the promoter or gene regions, and sometimes excision from the target sites [32]. TEs could regulate gene expression in spatial and temporal patterns or renovate the function adapted to developmental and environmental stimuli [33]. The bronze locus (bz) in maize is a structural gene for anthocyanin pigment biosynthesis in the aleurone layer of the endosperm. The Ds transposon insertion upstream of the transcription start sites and the footprint after transposon excision from the second exon resulted in gene expression changes and enzyme activity [34]. The stiff1 gene associated with the stalk strength in maize harbors a 27.2-kb Copia in the promoter, which represses the gene expression and, in turn, activates several key genes in cellulose and lignin synthesis, leading to a stronger stalk strength [35]. Cloning of the stiff1 allele with a transposon insertion provides an efficient method to improve maize stalk strength through marker-assisted selection. Another example illustrating that a transposon plays an important role in maize improvement is teosinte branched 1 (tb1). A Hopscotch transposon inserted in the regulatory region of tb1 enhances the host gene expression and represses the branch growth, partially explaining the increased apical dominance in the maize domestication. Transposons could not only create genetic diversity but also provide an essential source of functional variation during plant evolution and adaptation [12]. It seems that DNA transposons are mainly found in gene regions, whereas LTR retrotransposons are inserted in intergenic regions [31]. However, we identified both DNA transposons and LTR retrotransposons (Copia and Gypsy) inserted in the gene regions of SAT1, SAT3 and SAT4 (Figure 4). Transposition of an element into a gene often leads to inactivation of that locus due to the large spatial barrier against transcription or translation interruption. Whether the TE insertion into the SATs might affect gene expression or translation needs further experimental validation (Figure 4). The waxy (Wx) gene responsible for amylose biosynthesis is an excellent system for studying the effects of mutation, such as transposon insertion due to the easily scored waxy kernel phenotype. Three polymorphic alleles containing transposons in introns were identified. However, the transposon insertions interrupt long-range splice site recognition, leading to novel Wx transcripts because of different exons’ skipping. The Wx gene expression level was downregulated by approximately 10 times in mutants compared with wild types [36]. It is known that transposons are the targets of DNA methylation, mainly in order to silence their activities and inhibit subsequent proliferation. The methylation level in gene regions near transposon-flanking regions may directly impact the gene expression and prevent aberrant transcripts [37]. The cytosine methylation in the ZmCCT promoter was elevated with a CACTA-like transposon insertion, which dramatically reduced the ZmCCT transcription and photoperiod sensitivity [38].

### 4.3. Structural Variation of the SAT Family in NAMs

The NAM founder lines were sequenced using PacBio long reads and integrated Illumina short reads and optical maps, generating the high-quality assembled genomes. The B73-Ab10 genome containing an abnormal chromosome 10 was assembled into only 63 contigs with gapless chromosomes of 3, 9 and Ab10 [39]. The improved genomes will accelerate the identification of transposable elements, DNA rearrangement, methylation status and gene expression level [40]. It is known that these lines were selected to maximize the genetic diversity given the study of the association panel of 302 lines from the world [41]. Furthermore, the NAM founders enabled a higher power to dissect the QTLs (Quantitative Trait Loci) /genes contributing to phenotypic variation through joint linkage association analysis [14]. The kernel weight and composition were evaluated in a diverse panel, where teosintes and landraces had lower carbohydrate and higher protein content than the NAM parental lines, and NAM and landrace lines were significantly greater in kernel weight than teosinte lines [42]. Investigation of the phenotypic attributes including the kernel characteristics, kernel composition and metabolomic profiles in NAM inbred lines and their hybrids with B73 [30,42,43] revealed an extensive variation, providing a valuable source of new alleles for improving modern maize breeding. Our analysis identified that Ki11 seems to be an active genome with three transposon insertion events in the SAT gene family (Figure 4). The result is consistent with another dataset where only 32% of the assembled optical map in Ki11 could be mapped to the B73 reference compared with 39% in W22 [9]. There are abundant structural variations linked to phenotypic traits, especially agronomic variations that have not been explored. It was reported that the BARE-1 retrotransposon constituted a major and active component in the wild barley of *Hordeum*. The copy number was positively correlated with genome size and habitat aridity [44]. The transposon insertion polymorphism was also observed in rice, which explains 14% of sequence differences between subspecies indica and japonica. In addition, more than 10% of transposon insertions occurred in gene regions, representing a valuable source of genetic variation that may interrupt host gene expression or create novel transcripts [45].

## Figures and Tables

**Figure 1 genes-12-00437-f001:**
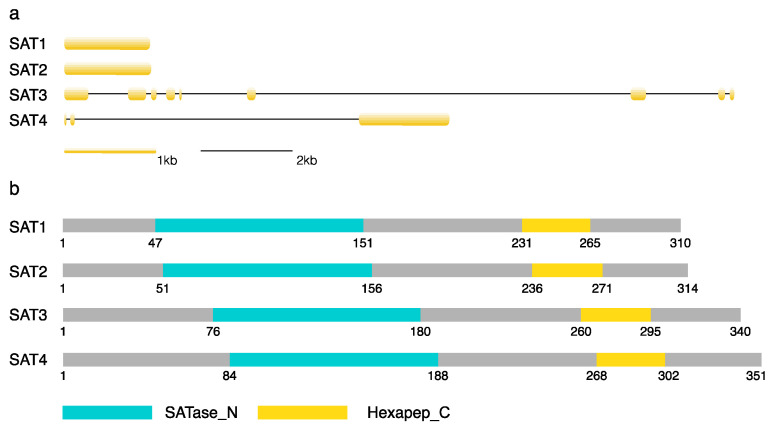
Gene structures and protein motifs for serine acetyltransferase (SAT) gene family in B73. (**a**) The exons are labeled in orange boxes and introns are illustrated by black lines. (**b**) The motif of serine acetyltransferase (SATase_N) at the N-terminal is colored in blue and the hexapeptide repeat domain (Hexapep_C) at the C-terminal is in yellow. Other protein regions are represented in grey boxes. The gene IDs are as follows: SAT1—Zm00001d011735; SAT2—Zm00001d028154; SAT3—Zm00001d027536; STA4—Zm00001d038737.

**Figure 2 genes-12-00437-f002:**
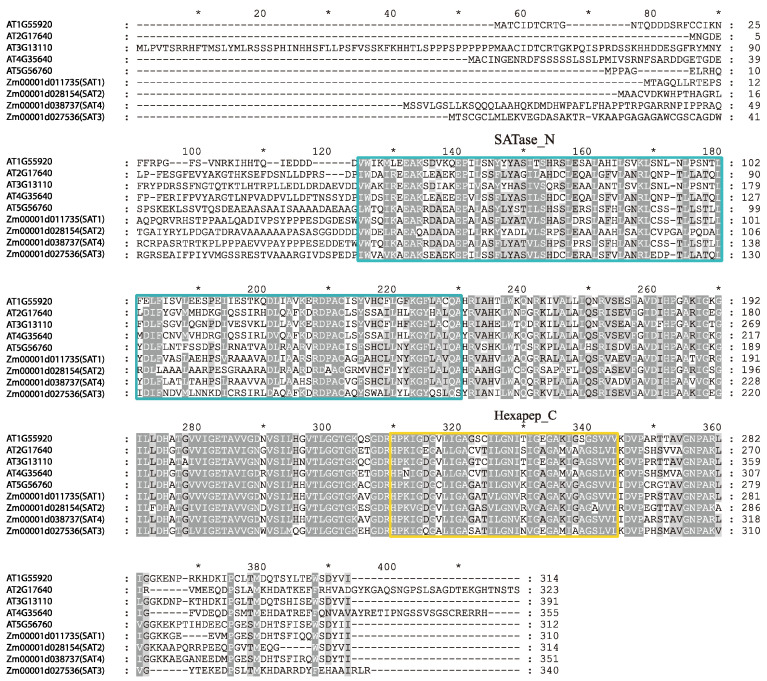
Amino acid sequence alignment of SAT gene family for B73 and *Arabidopsis*. The motifs of serine acetyltransferase (SATase_N) at the N-terminal and the C-terminal hexapeptide repeat domain (Hexapep_C) are highlighted with the blue and yellow boxes, respectively. The conserved amino acids are labeled as asterisks on top.

**Figure 3 genes-12-00437-f003:**
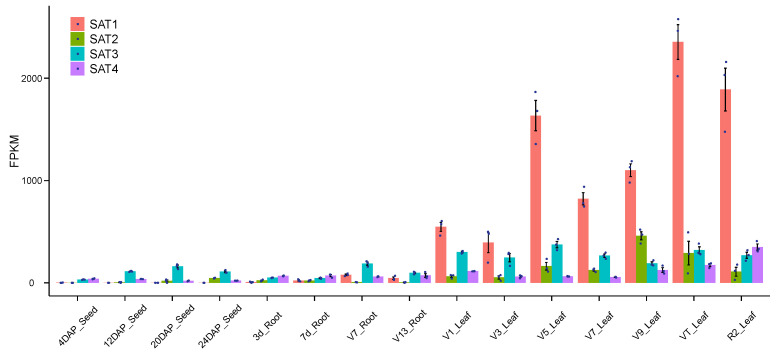
Quantification of SAT gene expression at developmental stages and different tissues in B73. The *x*-axis shows the sample names and the *y*-axis exhibits the fragments per kilobase of gene model per million reads mapped (FPKM) value of gene expression. The growth stages are represented by day after pollination (DAP), collar of 1st leaf visible (V1), collar of 3rd leaf visible (V3), collar of 5th leaf visible (V5), collar of 7th leaf visible (V7), collar of 9th leaf visible (V9), VT Tassel (all branches visible, no silks) and R2 (kernels are watery blisters with clear fluid). The raw data were generated by the Kaeppler lab using three biological replicates. The averaged FPKM value was downloaded from the paper of Stelpflug et al. [16] with standard errors.

**Figure 4 genes-12-00437-f004:**
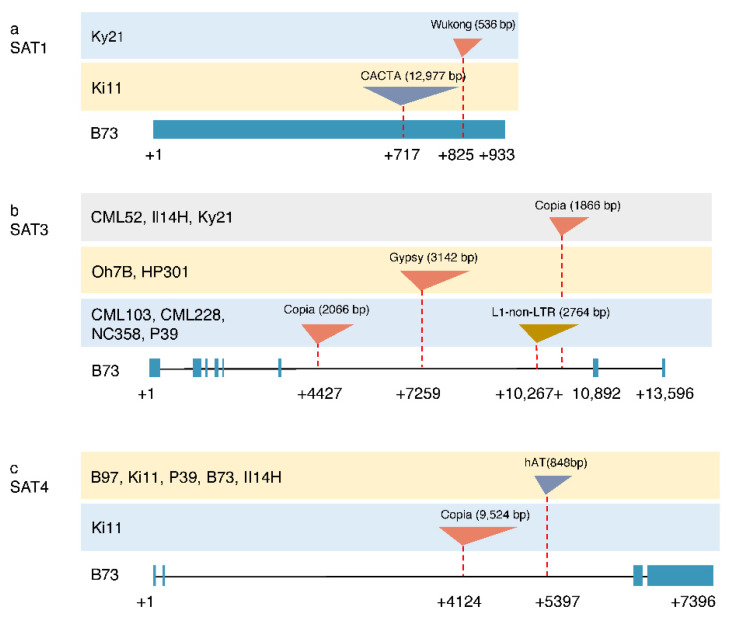
Transposon insertions in SAT1, SAT2 and SAT4 for nested association mapping (NAM) founders. (**a**) Transposon insertion in the SAT1 gene. (**b**) Transposon insertion in the SAT3 gene. (**c**) Transposon insertion in the SAT4 gene. The bottom line is the B73 reference, including exons with blue boxes and introns with black lines. The transposon type is colored by different triangles. SAT2 is skipped given no transposon insertion. Ky21, Ki11, CML52, Il14H, Oh7B, HP301, CML103, CML228, NC358, P39 and B97 represent the maize inbred lines of NAM founders.

**Figure 5 genes-12-00437-f005:**
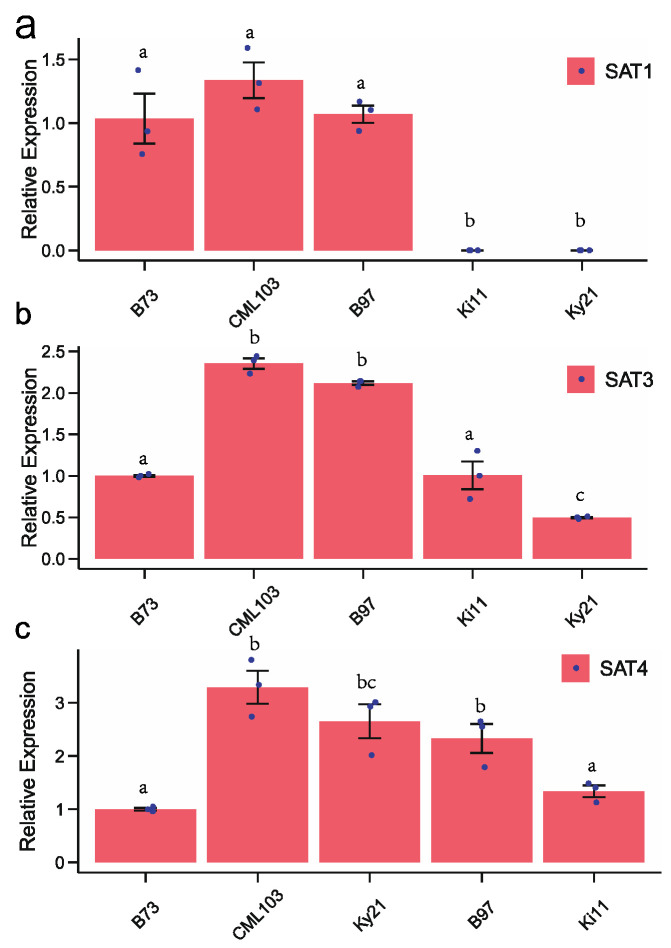
Quantitative real-time PCR for SAT gene expression. (**a**) SAT1 expression. (**b**) SAT3 expression. (**c**) SAT4 expression in B73, B97, CML109, Ki11 and Ky21. An ANOVA test was conducted to assess the statistical difference (*p*-value of 0.05). The black bars show the ± standard errors. “a”, “b”, “c” on top of each bar indicate statistics groups.

**Figure 6 genes-12-00437-f006:**
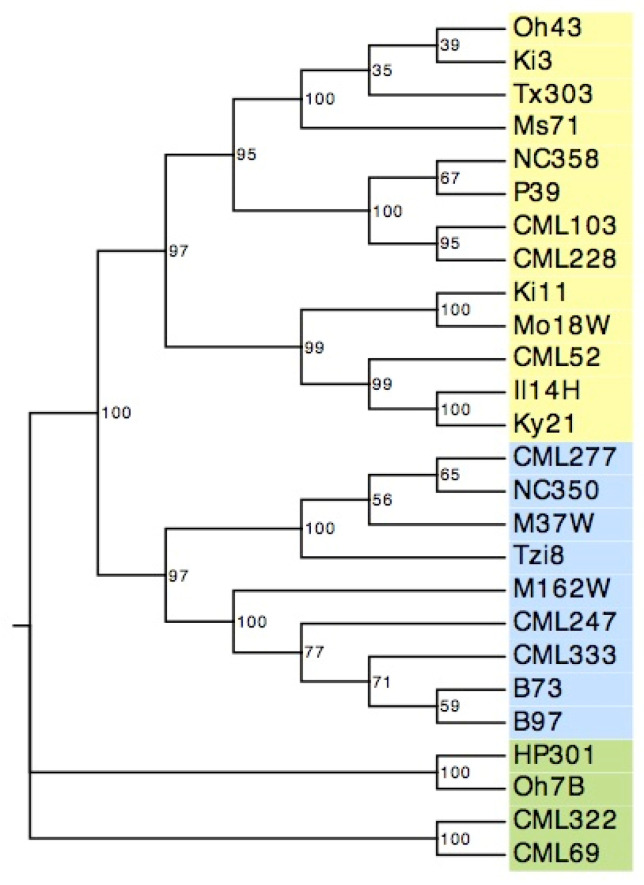
A phylogenetic tree based on SAT genetic distance for NAM founders. The evolutionary tree is inferred using the UPGMA method. The bootstrap consensus tree is inferred from 1000 replicates. The NAM founders are divided into three groups, labeled with yellow, blue and green. Oh43, Ki3, Tx303, Ms71, NC358, P39, CML103, CML228, Ki11, Mo18W, CML52, Il14H, Ky21, CML277, NC350, M37W, Tzi8, M162W, CML247, CML333, B97, HP301, Oh7B, CML322 and CML69 represent the maize inbred lines of 25 NAM founders.

**Figure 7 genes-12-00437-f007:**
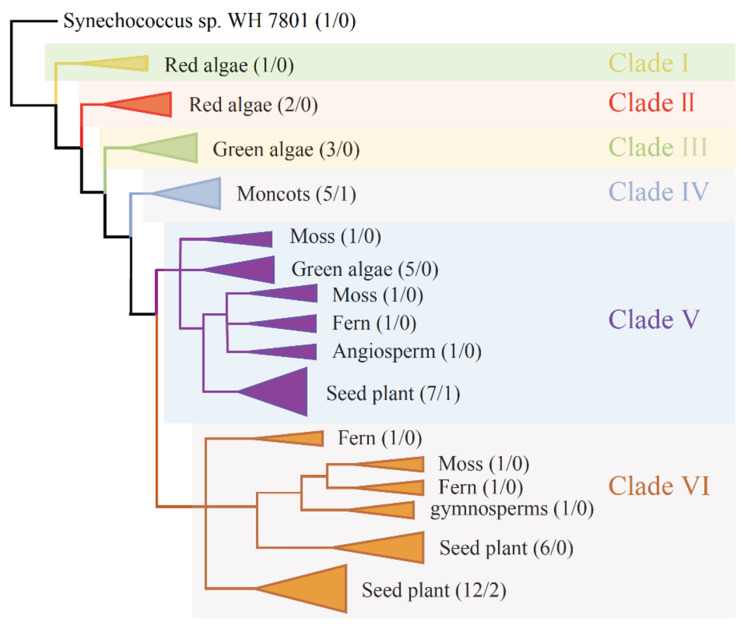
A phylogenetic tree of 50 SAT proteins from 16 species. The phylogenetic tree drawn by SAT protein sequences is divided into six clades (I, II, III, IV, V and VI) with different colors. Two numbers in parentheses represent the number of SAT proteins from all analyzed species and only from maize. The cyanobacteria of *Synechococcus* sp. WH 7803 is used as a root.

**Table 1 genes-12-00437-t001:** Features of SAT members including gene locus, gene size and molecular size.

Name	Genbank ID	Gene Locus	Gene Size (bp)	Exon Number	CDS Length (bp)	Protein (aa)	Molecular Weight (kDa)
SAT1	Zm00001d011735	chr8:159516273-159517206	933	1	933	310	32.5
SAT2	Zm00001d028154	chr1:24198740-24199685	945	1	945	314	32.4
SAT3	Zm00001d027536	chr1:7652686-7639117	13,596	9	1023	340	36.4
SAT4	Zm00001d038737	chr6:170865665-170873061	7396	3	1056	351	37.4

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
