# Peer review of "Genetic Variation of the Serine Acetyltransferase Gene Family for Sulfur Assimilation in Maize"

_genes, 2021, doi:10.3390/genes12030437_

Round 1

Reviewer 1 Report

Manuscript entitled as “Genetic variation of the serine acetyltransferase gene family for sulfur assimilation in maize” improved lot compared to last submission.

Please follow some minor corrections to improve the manuscript.

Line 19:  change to ‘insertions’ instead of inserted

Line 139: Please re write the following sentence “The sixth leaves at the R3 stage were sampled and frozen in -80 degree liquid nitrogen”. It is better way mention is  samples frozen in liquid nitrogen later stored at -80 degree,  

Line 197:  better to mention ‘low expression’ than ‘lowly expressed’

Author Response

Manuscript entitled as “Genetic variation of the serine acetyltransferase gene family for sulfur assimilation in maize” improved lot compared to last submission.

Response:Thank you. We are very honored to get your approval. The comments are valuable and very critical to improve our manuscript.

Please follow some minor corrections to improve the manuscript.

Line 19:  change to ‘insertions’ instead of inserted

Response:Thank you. We have revised it.

Line 139: Please re write the following sentence “The sixth leaves at the R3 stage were sampledand frozen in -80 degree liquid nitrogen”. It is better way mention is samples frozen in liquid nitrogen later stored at -80 degsree.

Response: Thank you. We polished the sentence accordingly.

Line 197:  better to mention ‘low expression’ than ‘lowly expressed’.

Response: Thank you. We have revised it.

Reviewer 2 Report

The authors make use of the existing databases to investigate the genetic variation, gene expression and gene evolution of SAT gene family, which is an essential enzyme controlling cysteine biosynthesis. Generally speaking, the manuscript is well written and easy to follow, and the analyses were solid. There are some minor comments though.

  • Line 19, spell out ‘TE’ for the first time in the summary.
  • Sulfate or sulphate are the same thing for different spelling, please keep just using one consistently throughout the manuscript, such as line 41, 43 and some others in the text.
  • Line 69, spell out TE for the first time. Also according to the examples the authors give in line 70-75, the statement of line 68-70 (‘There are …silencing signals’) should specify that these examples are for different crops or plant species.
  • Line 76, the authors may add the specific 25 lines in bracket. Also spell out ‘NAM’ for the first time in the introduction.
  • From the Results section, the authors did the polygenetic tree/Evolutionary analyses of SAT genes but did not mention it in the M&M. I would suggest change 2.4 Date source to 2.5, and add the polygenetic tree analyses as 2.4. Line 260-267 from Results should be moved here (2.4) too, they are not results, they should belong to M&M.
  • Line 206-210, it is repeated as in M&M, and they are not part of Results, should be removed.
  • I would suggest the authors keep colors consistent as in Fig 1 and Fig 2. They use blue and yellow to represent SATase_N and Hexapep_C, respectively, it will be nice to have the same color box in Fig 2 to represent the same thing.
  • In Figure 5 caption, how much is the standard error? One standard error or two in the plot?
  • Can the authors add citation for line 355-358?
  • I am not sure I understand how the sentence ‘Maize that originally requires … temperate regions’ in line 388-389 is related to the statements before and after.

Author Response

The authors make use of the existing databases to investigate the genetic variation, gene expression and gene evolution of SAT gene family, which is an essential enzyme controlling cysteine biosynthesis. Generally speaking, the manuscript is well written and easy to follow, and the analyses were solid. There are some minor comments though.

Response: Thank you for your comments and putting a lot of time in reviewing our manuscript. All suggestions and comments are invaluable to improve our manuscript.

Line 19, spell out ‘TE’ for the first time in the summary.

Response:We added it into the text: "transposon element (TE)".

,

Sulfate or sulphateare the same thing for different spelling, please keep just using one consistently throughout the manuscript, such as line 41, 43 and some others in the text.

Response: Thank you. We corrected them and kept the word of “sulfate” consistent in the text.

Line 69, spell out TE for the first time. Also according to the examples the authors give in line 70-75, the statement of line 68-70 (‘There are …silencing signals’) should specify that these examples are for different crops or plant species.

Response: Thank you. We added the spelling for TE. We also specified that these examples are for different crops or plant species.

Line 76, the authors may add the specific 25 lines in bracket. Also spell out ‘NAM’ for the first time in the introduction.

Response: Thank you. We added such information in the manuscript. “NAM” was also spelled in Introduction.

From the Results section, the authors did the polygenetic tree/Evolutionary analyses of SAT genes but did not mention it in the M&M. I would suggest change 2.4 Date source to 2.5, and add the polygenetic tree analysesas 2.4. Line 260-267 from Results should be moved here (2.4) too, they are not results, they should belong to M&M.

Response: Thank you for the good suggestions. We added phylogenetic tree analysis as 2.4 into the M&M and the redundant information in Results was deleted.

2.4. Phylogenetic tree analyses

    To study the SAT gene evolution, we performed the phylogenetic analysis using the identified 50 SAT proteins across cyanobacteria to seed plants, including a cyanobacte-rium of Synechococcus sp. WH 7803, two species of Galdieria sulphuraria, Chondrus crispus from red algae, three species of Micromonas sp. RCC299, Micromonas pusilla CCMP1545, and Chlamydomonas reinhardtii in green algae, one species of Mpolymorpha in moss, a fern of Selaginella moellendorffii, a gymnosperm of Picea abies, a basal angiosperm of Amborella trichopoda, four species (Brachypodium distachyon, sorghum, rice and maize) in monocots, and two species (Vitis vinifera, Arabidopsis) in dicots [22].

Line 206-210, it is repeated as in M&M, and they are not part of Results, should be removed.

Response: Thanks. We've moved the part into methods.

I would suggest the authors keep colors consistent as in Fig 1 and Fig 2. They use blue and yellow to represent SATase_N and Hexapep_C, respectively, it will be nice to have the same color box in Fig 2 to represent the same thing.

Response: The nice suggestion was taken, and the same color pattern was used in Fig 1 and Fig 2.

In Figure 5 caption, how much is the standard error? One standard error or two in the plot?

Response: The plot showed two standard errors. We added the sign of ± standard errors in the legend that could explain the meaning of standard error.

Can the authors add citation for line 355-358?

Response: Thanks. We added two related citations.

I am not sure I understand how the sentence ‘Maize that originally requires … temperate regions’ in line 388-389 is related to the statements before and after.

Response:Thank you. This sentence does not have any connection with our statement. We deleted the sentence.

This manuscript is a resubmission of an earlier submission. The following is a list of the peer review reports and author responses from that submission.

Round 1

Reviewer 1 Report

The authors have utilized published sequence data of the maize genome to investigate the genetic features of SAT gene families in terms of structural variance, gene expression and phylogeny. While no new data was used, the study is solid and made a broad review on several aspects that gave further information on the four SAT genes.

Some comments are given:

1. The English can be improved. Quite a few typos are found.

Some to mention

  • ln 280: The five clades -> the five clades
  • ln 285: recent evolved -> recently evolved
  • ln 292: , Two numbers -> , Two ...

2. The characteristics of SAT genes may be indirectly compared in other neighbor plant genomes for validation.

3. The description of data used in Section 2.1 should be further clarified. How were the SAT genes found in Maize. Were no SAT genes previously annotated in Maize?

Reviewer 2 Report

The manuscript entitled as “Genetic variation of the serine acetyltransferase gene family for sulfur assimilation in maize” tried to compile the SAT genes and compiled well. However, RNA sequencing data were retrieved from Kaeppler lab and SAT genes may be required for validation by Real-time PCR, this manuscript having still gaps to consider for the publication.

Some more minor comments as follows.

Line no13: please change to Genome-wide characterized Instead of ‘We genome-widely characterized’.

Line no 16: please change to Recently instead of ‘Thanks to the recent’

Line 62: please remove the words ‘Thanks to the’

Line 123: please correct to downloaded

Line 139: please correct the sentence “They were located on chromosome 1, 6 and 8, respectively”. It has to be either SAT1 and 2 on Chromosome 1 and SAT3 and SAT4  on chromosome 6 and 8, respectively or They were located on chromosome 1,1, 6 and 8, respectively.

Line 154: in fig1 b please correct to Hexapep_C instead of Hexapep at yellow colour labelling.

Line 177: correct to replicates instead of duplicates.